# Self-care interventions to reduce, prevent or monitor physical disability in those affected by leprosy: Protocol for a systematic review

Matthew Willis[1,2]*, Alexandra Asboeck[3], Bianca B. C. Amorim[4], Patricia D. Deps[4], Anil Fastenau[1,2,5], Melanie Boeckmann[1]

1 Department of Global Health, Institute of Public Health and Nursing Research, University of Bremen, Bremen, Germany, 2 Marie Adelaide Leprosy Center (MALC), Karachi, Pakistan, 3 School of Medicine and Health, Technical University of Munich (TUM), Munich, Germany, 4 Department of Social Medicine, Federal University of Espírito Santo, Vitória, Espírito Santo, Brazil, 5 German Leprosy and Tuberculosis Relief Association (GLRA/DAHW), Wurzburg, Germany

* willis@uni-bremen.de

## Abstract

### Background

Leprosy, also known as Hansen's disease, is a chronic infectious neglected tropical disease. Leprosy can cause extensive immune mediated nerve damage which can persist post cure of infection with multidrug therapy, leaving a significant burden of physical disability among those affected by leprosy. Leprosy can also cause irreversible damage to the eyes and eventual blindness or visual disability. One means of preventing, monitoring and managing the disability as a result of leprosy is through self-care. Self-care is the practice of patients undertaking activities to improve or maintain their own health with or without the support of a healthcare professional, and has been a part of World Health Organization Zero leprosy strategies for many years. This review will seek to identify what self-care interventions exist for leprosy affected people and will map these interventions to the transtheoretical model of behaviour change to identify which level of the model current interventions target. Sub questions will explore what information is provided regarding the cost of the interventions and additionally explore what eye, feet and hand specific self-care advice is given by interventions.

### Methods

We will search for literature in PubMed, Medline ALL, Web of science Core Collection, Embase, LILACS and SciELO. Studies that encompass a self-care intervention under the definition provided by the World Health Organization targeting those affected by leprosy will be eligible for inclusion. No language restriction will be applied. Data will first be deduplicated using endnote before the final deduplicated

**Data availability statement:** No datasets were generated or analysed during the current study. All relevant data from this study will be made available upon study completion.

**Funding:** The author(s) received no specific funding for this work.

**Competing interests:** The authors have declared that no competing interests exist.

**Abbreviations:** WHO, World Health Organisation; NTDs, Neglected Tropical Diseases; TTM, Transtheoretical model of behaviour change; PICOS, patient/population, intervention, comparison and outcomes; PRISMA, Preferred Reporting Items for Systematic reviews and Meta-Analyses; CASP, Critical appraisal and skills checklist; MeSH, Medical Subject Headings.

search results are uploaded to Rayyan for screening by the research team. We will apply a blinded screening whereby if at least one screener identifies a paper as potentially relevant it will proceed to full text screening, this will be carried out independently by at least two of the authors, in this case of disagreement a final decision will be made by the senior author. Quantitative, qualitative and mixed method studies will be eligible for inclusion. Studies will be assessed for bias using CASP checklists. Findings will be reported in narrative synthesis and as a visual representation of levels of the transtheoretical model addressed in interventions. Start date: 11 August 2025. End date: 30 November 2025.

## Discussion

The findings from this review will document existing self-care interventions for those affected by leprosy, mapping the individual interventions to the stage of the transtheoretical model of behaviour change that they address. Information will also be presented regarding implementation costs of the interventions and what specific advice for hands, feet and eyes is given. Highlighting promising interventions can support clinicians in low-resource settings in advising their patients on what to prioritize in their self-care.

## PROSPERO registration no.

CRD42025649623.

## Background

Leprosy, also known as Hansen's disease, is a chronic infectious neglected tropical disease (NTD) caused by *Mycobacterium leprae* and *mycobacterium lepromatosis* [1]. Leprosy is transmitted when droplets from an untreated leprosy patient spread to susceptible individuals; the disease has a low infectivity and as such requires months of prolonged contact for infection to occur. The disease primarily affects the skin, nerves, eyes and mucous membranes. The host immune response against the bacteria then results in immune driven nerve damage which can lead to injuries such as cuts and burns due to the loss of sensation [2]. This is additionally relevant in ocular leprosy as facial nerve involvement can cause corneal anaesthesia leading to neuropathic keratopathies and lagophthalmos [3,4]. If left untreated this can lead to severe visual impairment and eventual blindness.

Although multidrug therapy (MDT) interrupts the spread and "cures" patients microbiologically of infections, it does not remove the risk of further deterioration in neurological and ocular disability [2,5]. As such from this point on this manuscript will refer to "people affected by leprosy" as this includes those who have been successfully cured of active infection by MDT. Those affected by leprosy require long term management post-treatment to monitor nerve and visual function, requiring a significant level of healthcare resources.

One relatively low resource means of monitoring and controlling the consequences of the nerve damage is through self-care [6]. Self-care encompasses a broad spectrum of behaviours and routines that individuals engage in to enhance their health and prevent disability [7]. The World Health Organization (WHO) defines, self-care as "the ability of individuals, families and communities to promote health, prevent disease, maintain health, and cope with illness and disability with or without the support of a healthcare provider." [8]. Analysing the idea of self-care in leprosy presents certain challenges. Self-care overlaps with related concepts such as self-management, self-monitoring, self-treatment, self-efficacy, and peer support, all of which are interventions that share similar goals of patient empowerment and self-guided care [9].

Self-care in leprosy is primarily aimed at reducing future disability from the nerve damage. Wound and skin care is vitally important in those affected by leprosy, particularly those with already established nerve damage [6]. Cleaning and dressing wounds prevents the repeated damage of areas and allows healing to occur. Additionally, leprosy affected people must take extra precautions due to temperature sensation loss, such as using kitchen gloves to handle hot pans and testing the temperature of water before bathing [10]. Regular exercise and stretching are also essential to maintain muscle function due to potential nerve damage leading to wasting [11].

As part of the management of the ocular manifestations such as dry eye, loss of corneal sensation and lagophthalmos, regular self-eye-care is recommended as part of daily life in those affected by leprosy [12]. These recommendations are designed to reduce ocular related disability by reducing the effect of the ocular complications of leprosy. Strategies involve the encouragement of eye protection such as glasses and shawls that can protect the eye from dust and grit. Additionally, they recommend careful cleaning of the eye to remove debris build up and in those with lagophthalmos it is important to gently cover or tape the eyes at night to prevent exposure of the cornea during sleep [12]. Self-care can also serve as a means to prevent disability by serving as a trigger for healthcare attention, for example by self-examination for red eyes [12].

## Behaviour change as an explanatory framework for the uptake of self-care

The Transtheoretical model of behaviour change (TTM) [13] serves a framework with which to analyse the individual barriers to behaviour change. The TTM maps behaviour across 5 main levels: 1) precontemplation, 2) contemplation, 3) preparation, 4) action, and 5) maintenance, and a subsequent possible phase of relapse. Self-care is an individual level intervention with which the disability burden of leprosy can be addressed; however, many psychological and practical barriers to the practice of self-care exist [14]. Previous research has identified barriers at different levels of the TTM. Within precontemplation, poor knowledge surrounding self-care may act as a barrier [15,16]. Stigma associated with leprosy [17,18] may be theorised to keep patients in the contemplation stage of the TTM as participating in self-care may disclose their disease status. The consequences of leprosy itself may also present a barrier once patients reach the action phase due to limb disability impairing the ability to dress and bandage wounds or perform self-examination, furthermore the worsening or continued impact of disabilities may serve as a barrier within the maintenance phase [2]. Additionally, a lack of clean water or appropriate equipment is present in many endemic communities [19,20].

Mapping current self-care interventions to the level(s) of the TTM they target can provide an inventory of interventions for different stages of the TTM process. Although self-care interventions for leprosy have been previously investigated by two scoping reviews [6,21], these reviews investigated different themes. Casado et al. [21] investigated community programmes more broadly and Ilozumba and Lilford [6] focussed on ulcers rather than on disability. This review will therefore seek to document current self-care interventions addressing physical disability for those affected by leprosy and analyse how they address behaviour change within the TTM. Highlighting promising interventions can support clinicians in low-resource settings in advising their patients on what to prioritize in their self-care.

## Aims

This study aims to

1. Describe the contents and effectiveness of self-care interventions addressing physical disability in leprosy and evaluate which level of the TTM the identified interventions address

2. Summarise what specific self-care advice is provided for the hands, feet and eyes of those affected by leprosy

3. Summarise information available about the cost to implement the interventions in the provided setting.

## Methods

### Design

A systematic review, following the PRISMA guidelines [22], will be carried out to synthesise evidence on what self-care interventions for the monitoring, prevention and reduction of disability exist for people affected by leprosy, and to collate available information about which of these interventions include eye, hand and feet specific advice and information about cost to implement the intervention. The PRISMA – P checklist is attached as S1 File and the section in which each point is addressed is bracketed at the end of the relevant checklist item. Start date: 11 August 2025. End date: 30 November 2025. Record screening complete by October 30th$^{st}$. Data extraction will be completed November 15$^{th}$. Review finished November 30th.

### Search strategy

An automated search will be completed in PubMed, Medline ALL, Web of science Core Collection, Embase, LILACS and SciELO. The MeSh terms for PubMed will be generated based on the PICOS criteria outlined in eligibility criteria and then translated to the other databases.

The search will encompass the following broad categories

1. Leprosy

2. Self-care

We will search for studies that address self-care interventions as defined by the WHO for those affected by leprosy, no geographical or date limit will be imposed.

Terms describing leprosy – S1: leprosy OR Hansen's disease

Terms describing Self-care - S2: Self-Care OR self-management OR self-treatment OR self-treatment OR self-efficacy OR peer support OR patient empower* OR Self-examination OR self-guided care

S3: S1 AND S2.

The search process will be presented in a PRISMA flow chart to indicate how many hits were retrieved from each included database and the inclusion/exclusion justification for full text screening.

Proposed search strategies are provided in detail in S2 File.

### Eligibility criteria

There will be no restrictions on publication dates, language, or geographical location. Where possible the English language abstract will be reviewed, if the manuscript or abstract is in a language other than that spoken by the included reviewers it will be translated to English and reviewed. Studies will only be included where a self- care intervention, using the WHO definition of self-care, aimed to prevent, monitor or reduce leprosy related physical disability. The definition of self-care for this manuscript was the WHO definition of "the ability of individuals, families and communities to promote health, prevent disease, maintain health, and cope with illness and disability with or without the support of a healthcare provider." [8].

Original peer reviewed research using either qualitative, quantitative, or mixed method approaches will be eligible for inclusion. The PICOS criteria [23] used for this study will be:

Population – People affected by leprosy;

Intervention – a self-care intervention designed to reduce the impact of, monitor the progression of or prevent the occurrence of a physical disability associated with leprosy;

Comparison – what did the researchers conclude regarding effectiveness, applicability and cost to implement such interventions. Which level(s) of the TTM do the identified interventions target;

Outcome – reported results of intervention in regard to monitoring, preventing or reducing disability in those affected by leprosy, cost to implement intervention and eye, feet and hand specific advice provided;

Study type – Original, peer-reviewed research encompassing qualitative, quantitative and mixed-methods studies. Meta-analysis and review papers will be excluded.

## Data collection and management

Data will initially be maintained and managed using endnote referencing software. Duplicates will be manually excluded using endnote before transferring the articles to Rayyan [24]. At least two authors will then complete both title and abstract screening using Rayyan's "blind" function, whereby if at least one screener identifies a paper as potentially relevant it will proceed to full text screening. Full text screening will then be carried out individually by at least two authors before discussing discrepancies, after which if consensus cannot be reached it will be mediated by the senior author. The included articles will then be then reviewed independently by at least two authors, and information collated and tabulated on a standard form for all papers attached as S3 File.

## Data extraction and analysis

Data extraction will include setting, sample, study design, self-care intervention/techniques studied, measures provided by researchers, effects, this is summarised in S3 File). Cost of intervention if available will be summarised in a table that will be attached as an appendix attached as S4 File, specific eye, feet and hand advice will be summarised in a separate appendix attached as S5 File. This will initially be completed independently by two authors before a consensus is reached by all authors following discussion on what data to include in the table. Interventions will then be thematically analysed and mapped onto the TTM after the results have been analysed. This will include what level(s) of the TTM are addressed or targeted by the included intervention and how the intervention identified does this.

## Risk of bias

In keeping with PRISMA guidelines each paper will be screened for independent bias. CASP checklists [25] will be used to screen for any potential bias, this will be carried out by at least two authors before discrepancies will be addressed by the senior author.

## Dissemination plan

The final manuscript will be submitted for review and publication in leading journals of NTDs and global health. Findings relevant to future practice in leprosy programmes will be disseminated to relevant practitioners through its publication and/or submission to conferences.

## Discussion

The findings from this review will document existing self-care interventions for those affected by leprosy, mapping the interventions to stage of the transtheoretical model of behaviour change. Information will also be presented regarding

the cost of the interventions to implement and what eye-specific advice is given. Data relevant to the cost to implement interventions will be summarised in order to help policy makers and researchers decide if an intervention is suitable for their locality. Additionally, eye, feet and hand specific advice will be summarised due to the specific nature of ocular, hand and foot disability that leprosy can cause and the lack of a previous paper summarising the information available. The findings will help formulate interventions that can be designed to encourage or improve self-care practices among those affected by leprosy. Since leprosy affected patients experience a relapsing and remitting pattern of neurological disability as a result of the nerve damage self-care interventions are a vital component in Zero leprosy strategies and they allow for individual and community level monitoring and management of disability in an integrated manner post elimination of transmission of leprosy.

## Supporting information

**S1 File. PRISMA-P-checklist.**
(DOCX)

**S2 File. Sample search strategy_ENG_ESP_POR.**
(DOCX)

**S3 File. General data extraction table.**
(DOCX)

**S4 File. Cost of interventions table.**
(DOCX)

**S5 File. Eye, Hand and feet data extraction table.**
(DOCX)

## Author contributions

**Conceptualization:** Matthew Willis, Alexandra Asboeck, Patricia D. Deps, Anil Fastenau, Melanie Boeckmann.

**Investigation:** Matthew Willis, Alexandra Asboeck.

**Methodology:** Matthew Willis, Bianca B. C. Amorim, Patricia D. Deps, Melanie Boeckmann.

**Project administration:** Matthew Willis.

**Supervision:** Melanie Boeckmann.

**Writing – original draft:** Matthew Willis, Alexandra Asboeck, Anil Fastenau, Melanie Boeckmann.

**Writing – review & editing:** Matthew Willis, Alexandra Asboeck, Bianca B. C. Amorim, Patricia D. Deps, Anil Fastenau, Melanie Boeckmann.

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
