## [Decision Letter · Decision Letter 0]

4 Aug 2025

Self-care interventions to reduce, prevent or monitor physical disability in those affected by leprosy: protocol for a systematic review

PONE-D-25-10635

Dear Dr. Willis,

We’re pleased to inform you that your manuscript has been judged scientifically suitable for publication and will be formally accepted for publication once it meets all outstanding technical requirements.

Kind regards,

José Luiz Fernandes Vieira

Academic Editor

PLOS ONE

Journal Requirements:

**Additional Editor Comments**

I have additional comments to improve your systematic review, which should be considered for inclusion in the final version of the manuscript:

Regarding the Background section, I recommend revising the sixth paragraph to enhance readers’ understanding of the Transtheoretical Model (TTM) stages, as only a few examples are currently provided. The last sentence of that paragraph should be moved to the following paragraph to strengthen the study’s justification.

All methods proposed by the authors align with established frameworks for conducting systematic reviews.

I recommend that the authors cleraly cite that meta-analyses and review articles will be excluded  from the study selection criteria and suggest including information on the types of lesions presented by the patients.

Reviewers' comments:

Reviewer's Responses to Questions

**Comments to the Author**

1. Does the manuscript provide a valid rationale for the proposed study, with clearly identified and justified research questions?

Reviewer #1: Yes

2. Is the protocol technically sound and planned in a manner that will lead to a meaningful outcome and allow testing the stated hypotheses?

Reviewer #1: Yes

3. Is the methodology feasible and described in sufficient detail to allow the work to be replicable?

Reviewer #1: Yes

4. Have the authors described where all data underlying the findings will be made available when the study is complete?

Reviewer #1: Yes

5. Is the manuscript presented in an intelligible fashion and written in standard English?

Reviewer #1: Yes

You may also provide optional suggestions and comments to authors that they might find helpful in planning their study.

Reviewer #1: The rationale for the study is clear and valid. The study addresses the significant burden of physical disability caused by leprosy, even after microbiological cure, and highlights the importance of self-care interventions as a low-resource strategy to prevent, monitor, and manage disability. The protocol is technically sound and designed to effectively achieve its aims. It outlines a systematic review methodology following PRISMA guidelines.

The Transtheoretical Model is a useful starting point for understanding self-care behaviors, especially in personalized care planning. However, its effectiveness depends on how well it is adapted to include social, cultural, and systemic factors. As a standalone framework, it may fall short in addressing the broader determinants of self-care behavior.

**Do you want your identity to be public for this peer review?** For information about this choice, including consent withdrawal, please see our Privacy Policy

Reviewer #1: **Yes: ** Joydeepa Darlong

---

## [Editor Report · Acceptance letter]

PONE-D-25-10635

PLOS ONE

Dear Dr. Willis,

I'm pleased to inform you that your manuscript has been deemed suitable for publication in PLOS ONE. Congratulations! Your manuscript is now being handed over to our production team.

Kind regards,

on behalf of

Dr. José Luiz Fernandes Vieira

Academic Editor

PLOS ONE